# Legumes or Meat? The Effectiveness of Recommendation Messages towards a Plant-Based Diet Depends on People’s Identification with Flexitarians

**DOI:** 10.3390/nu15010015

**Published:** 2022-12-21

**Authors:** Valentina Carfora, Patrizia Catellani

**Affiliations:** Department of Psychology, Catholic University of the Sacred Heart, 20123 Milan, Italy

**Keywords:** environmental message, persuasive communication, dynamic norm meat, legumes, addition, substitution

## Abstract

In the present research, we analyzed how to promote a plant-based diet by involving 428 volunteers in a 2-week mobile app intervention. We compared messages promoting the addition of legumes versus messages promoting the replacement of meat with legumes. Messages were either combined or not combined with dynamic norms (i.e., information that more and more people are enacting the behavior). We compared these messages with a control condition (i.e., no message intervention) and we also analyzed the moderation effect of receivers’ identification with flexitarians (i.e., people who occasionally eat animal products) and attitudes towards them. In the short term, addition messages increased legume consumption more than replacement messages, especially in people with a negative evaluation of flexitarians and low identification with them. In the long term, increased legume consumption was recorded only when addition messages were combined with dynamic norms. As for meat consumption, the replacement messages were more effective in reducing it in the short term than in the long term, especially in people with positive attitudes towards flexitarians. However, replacement messages combined with dynamic norms were more effective in the long term than in the short term. These results advance our comprehension of how to tailor dietary messages.

## 1. Introduction

Food production is one of the main factors related to current environmental problems [1,2,3]. Among food systems, the livestock industry is one of the most impacting sectors, contributing between 12% and 18% to global GHG emissions [4,5]. Nevertheless, promoting sustainable methods for the industry is not enough to reduce the environmental footprint of food production [6], given that we need a consumer transition toward more plant-based diets [6,7,8,9].

So far, Europeans still do not adopt a sustainable diet, with only 10% of consumers being flexitarians (i.e., they do not eat meat regularly), vegetarians (i.e., they do not eat meat and fish), or vegans (i.e., they do not eat animal products) [10,11]. However, consumers have started to select more and more sustainable foods and alternative sources of proteins (e.g., legumes, and other products that use legume-driven ingredients) [12,13,14].

To support this growing trend, some European multinational projects have focused on the importance of legumes by defining how they improve both consumers’ health and the environmental profile of European farming [15]. Legumes are edible seeds of the Fabaceae (Leguminosae) family. This family of products includes crops grown for seed (e.g., dry beans, dry peas, and lentils), fresh vegetables (e.g., green beans and green peas), and livestock forage (e.g., clover and alfalfa). Especially if consumed as meat alternatives, legumes represent healthy, high-protein, and nutrient-dense products that can prevent various chronic diseases (e.g., diabetes, bowel cancer, overweight, heart disease, and stroke), [16] and they provide various essential benefits to the ecosystem [17,18,19].

In the present study, we tested the impact of diverse messages promoting a switch from meat to plant-based diet, referring to the environmental benefits that may derive from such a switch. To achieve this aim, we assessed the persuasiveness of messages focused on one of two different dietary strategies. The first strategy consisted of inviting participants to add more legumes to their diet (i.e., *addition messages*). The second strategy encouraged participants to replace meat with legumes (i.e., *replacement messages*). We also assessed whether the persuasiveness of these messages would vary according to their being combined (or not) with informing participants that more and more people are enacting the suggested behavior (i.e., with reference to a dynamic descriptive norm). Finally, we assessed whether the various messages would be differently persuasive according to prior levels of participants’ identification with flexitarians and positive/negative attitudes towards them.

## 2. Theoretical Framework

### 2.1. How to Formulate Persuasive Messages to Promote the Consumption of Legumes

Numerous studies have tested the effectiveness of different types of messages to promote a change in eating behavior, such as messages focused on health benefits, on emotional components, or on the presentation of images (for a review on meat reduction, see [20]). Prior studies on the effects of messages promoting a change in eating behavior via a focus on environmental benefits found mixed results (e.g., [20,21,22,23,24,25,26,27,28]). These results suggest that the persuasiveness of these messages might depend on how they are formulated. So far, most researchers using these messages have indifferently leveraged three different dietary strategies, namely addition, replacement, and elimination [29]. *Addition* refers to the inclusion of a specific food in the diet, *replacement* regards the substitution of a food option with another one, and *elimination* relates to totally giving up a certain food. Only one study compared the effectiveness of addition and replacement messages with the aim of promoting a flexitarian diet, that is, a diet based on a reduced consumption of meat but not on its complete elimination [30]. The results of this study indicated that when people had never eaten plant-based meat, both addition and replacement messages increased positive attitude toward plant-based meat, and in turn willingness to pay for it. Instead, when people already ate plant-based meat, only replacement messages were effective. Evidently, people who have already added plant-based meat to their diet are more ready to progressively reduce their meat consumption

Prior research also demonstrated that food choices can be changed more easily by leveraging a descriptive social norm, that is, providing information on what others eat, than by leveraging an injunctive norm, which suggests what people should eat [31,32,33]. However, most previous studies have used *static descriptive norm* messages, which address the current state of the norm [34], and providing information on a static norm is usually effective when the promoted behavior is already common [35]. Communicating that only a minority of people performs a desirable behavior does not encourage conformity and can even backfire, as it can establish the norm of *not* engaging in the behavior [36]. 

The switch from meat to plant-based diet is not so widespread so far, even if it is an increasing trend. In this case, we can therefore focus on another type of social norm, namely a *trending minority norm* or *dynamic descriptive norm* [37,38,39]. Such a norm implies that increasingly more people are beginning to engage in a given behavior. Referring to dynamic norms has been recently introduced as a strategy to increase adherence to advisable but not (yet) common behaviors, motivate behavioral change, and lead individuals to anticipate a changing world [37,38]. In the case of the promotion of meat reduction, informing meat-eaters that more and more people are reducing their meat consumption increases their interest in behaving accordingly [39], also when combined with environmental messages [40], and even more so when the message source is a researcher [41]. However, other studies found no effect of dynamic norm information on participants’ interest in reducing meat consumption [42,43]. Given these mixed results and new evidence that the way the dynamic norm is communicated influences its effectiveness, much remains to be discovered about the potential of dynamic norms to change eating habits.

Importantly, a two-study paper compared the effect of a replacement message (i.e., encouraging readers to reduce their meat consumption) and an elimination message (i.e., encouraging readers to completely give up eating meat), both combined with information on the dynamic norm (i.e., informing that an increasing number of people is reducing meat [44]). Study 1, conducted with an online convenience sample, showed that both appeals initially increased participants’ favorable attitudes towards reducing meat consumption and the intention to eat less meat. However, only the replacement message decreased self-reported meat consumption. Study 2, conducted with a nationally representative sample, confirmed that both messages increased attitudes and intentions. Despite this, none of the treatments impacted meat consumption one month and two months after the end of the experiment. It is important to note that in this study, the authors did not test the addition messages, probably because they aimed at eliminating meat consumption instead of reducing it by adding more plant-based food.

In sum, prior results confirmed the effectiveness of addition and replacement messages in changing receivers’ intention toward more sustainable diets, also when combined with dynamic norms. However, prior research did not observe a significant effect of these messages on actual behavior. Precisely, the former did not include a measurement of participants’ food consumption changes [30]. The latter included this measurement but did not record any significant changes [44]. Therefore, it remains to be clarified whether behavioral change can be achieved with environmental messages and to what extent this is possible.

In this vein, in the present study, we verified the persuasiveness of a 14-day intervention comparing addition and replacement environmental messages, combined or not with the provision of information on dynamic norm. Precisely, given the inconsistency of prior results, we compared these four messages without hypothesizing any specific differences among them. Thus, we posed the following research question.

**Research Question 1** (**RQ1**). *To what degree are addition and replacement messages, with or without a reference to a dynamic norm, effective in influencing consumers’ diets?*

### 2.2. The Moderating Role of Attitude toward and Identification with Flexitarians

Food selection, and specifically the choice to eat or to not eat meat, is influenced by the groups to which one feels to belong or not to belong [45,46,47,48], given that people are likely to consciously or unconsciously prefer certain foods according to group values and norms (e.g., [49,50,51]). This is especially true when the pro-environmental outcomes of a dietary change can only be achieved by collective efforts [52]. Indeed, sustainable food choices are directly or indirectly affected by whether they are consistent with the reference group norms. That means people should clearly categorize the self in a group involved in the desired dietary change [53], so that these ingroup norms and goals (e.g., selecting sustainable diets) give group members’ actions direction and purpose.

In the context of the study of communication effects, we can expect identification with a social group to increase the effectiveness of messages that are consistent with the group norms and habits [54].

Considering that all our messages implicitly suggested that people are getting closer to the flexitarian group (i.e., social group who adopt a diet based on plant-based foods and reduced meat consumption), we expected that message effectiveness would vary based on the receivers’ pre-existing level of identification with flexitarians. Thus, people who identify with the flexitarian social group should be more willing to listen to messages promoting the eating habits of flexitarians, that is, the consumption of plant-based products and their introduction in the diet to replace meat. Therefore, we formulated the following hypothesis.

**Hypothesis 1a** **(H1a).***People with a strong identification with flexitarians are more persuaded by all message conditions, compared to control*.

People who do not identify with the flexitarian social group might be more resistant to messages promoting eating habits attributable to this group, and therefore, they should be more persuaded by messages promoting the consumption of legumes without referring to meat and dynamic norms.

**Hypothesis 1b** **(H1b).***People with a weak identification with flexitarians are more persuaded by addition messages (i.e., messages promoting the consumption of legumes without mentioning meat reduction), compared to the other conditions*.

Food choice is conditioned by the individual’s positive or negative attitude towards social groups characterized by peculiar food habits. Consistently, peoples’ prior attitude towards a social group influences the perception of messages promoting food habits that can be attributed to such groups. For example, the baseline attitude toward vegans influences the effect of messages focused on the vegan diet [55]. We can therefore expect people who positively evaluate flexitarians to be persuaded by messages suggesting the habits of the flexitarian group (i.e., reducing meat consumption and replacing it with plant foods).

**Hypothesis 2a** **(H2a).**
*People with a positive attitude toward flexitarians are more persuaded by replacement messages (i.e., messages promoting the consumption of legumes instead of meat), compared to control.*


Vice versa, people who negatively evaluate flexitarians are more persuaded by messages suggesting the addition of more planted-based foods to their diet, without mentioning meat replacement.

**Hypothesis 2b** **(Hb).**
*People with a negative attitude toward flexitarians are more persuaded by addition messages (i.e., messages promoting the consumption of legumes without mentioning meat reduction), compared to the other conditions.*


Third, we can expect an interaction between consumers’ identification with flexitarians and their prior attitude towards this group. Those who are not yet identified with flexitarians, but evaluate them positively, would be more persuaded by messages informing them how people are managing to adopt a flexitarian diet. This would be the case because if we know that more and more individuals are succeeding in taking on the eating habits of a positively valued group of people, then we should be motivated to try to do the same.

**Hypothesis 3a** **(H3b).**
*People with a positive attitude towards flexitarians and with a low identification with them are more persuaded by replacement messages with information on the dynamic norm than by the other conditions.*


Conversely, if people do not identify themselves as flexitarians and evaluate them negatively, they should be more persuaded by messages inviting them to add more legumes to the diet, without including a reference to the reduction in meat and the behavior of those who are approaching flexitarianism.

**Hypothesis 3b** **(H3b).**
*People with negative attitudes towards flexitarians and with a low identification with them are more persuaded by addition messages without information on the dynamic norm, compared to other conditions.*


To test our hypotheses, we evaluated the effectiveness of the different message interventions by observing the changes over time in self-reported consumption of legumes and meat.

Finally, we were interested in analyzing whether changes in participants’ consumption of legume and meat consumption after message exposure would remain stable over time, i.e., one month after the end of the message intervention. Over that period, all participants received no messages. As we could not control whether relevant other factors would intervene in this period (e.g., exposure to other environmental information), we decided to investigate this aspect with caution, without formulating a hypothesis regarding the long-term effect of the intervention. We compared the long-term effect of the intervention to a baseline level before the intervention took place and to the short-term effect immediately after the end of the intervention.

**Research Question 2** (**RQ2**). *Do changes in participants’ consumption of legumes and meat after message exposure remain stable one month after the end of the message intervention?*

In this way, we aimed at addressing one of the major limitations of most previous studies testing the effects of message interventions, i.e., not including a follow-up to test whether message effects are still present after the intervention ended (but see [44]).

## 3. Methods

### 3.1. Sample and Procedure

Using GPower 3.1, we conducted a sample size estimation considering a medium *Cohen’s d* effect size (ES = 0.15). With an *alpha* = 0.05, *power* = 0.90, *number of groups* = 5 (4 message conditions and control), *number of measurements* = 6 (self-reported consumption of meat and self-reported consumption of legumes at 3 time points), and *p* = 0.05, the projected sample size needed was approximately *N* = 428, with 85 participants per group.

In October 2021, Italian adults between 18 and 80 years of age were invited to participate by the students as volunteers in a study of the Catholic University of the Sacred Heart. The inclusion criteria were that participants should have no medical conditions and eat at least three portions of meat a week. At T1 (Time 1), participants completed the first questionnaire via the PsyMe app. (The PsyMe app is a free smartphone app developed by the Catholic University of the Sacred Heart and Pavia University, developed to support scientific research in the field of social psychology and artificial intelligence. The PsyMe app respects participants’ privacy and anonymity, thanks to the assignment of an anonymous code to each participant. The PsyMe app allows sending questionnaires, messages, and push notifications to remind message reading).

After completing the first questionnaire, participants included in the study were randomly assigned to one of the five different experimental conditions using an automatic randomization sequence. Participants who did not complete the entire questionnaire or failed control questions were excluded from the research. We invited participants until we reached the targeted sample size (*N* = 428; mean age = 34.73, *SD* = 6.80; age range: 18–80; F = 207; M = 216; undeclared = 2).

Then, every day at 9:30 and for 14 days (between T1 and T2), participants received one message focused on the environmental consequences of eating legumes (see the section below). At T2 (Time 2), that is, at the end of the 14-day intervention, 296 participants completed a second questionnaire on their evaluation of the messages and their weekly consumption of meat and legumes (mean age = 35.29, *SD* = 16.44; F = 156; M = 142; undeclared = 2). At T3 (Time 3), that is, one month after the end of the intervention period, 221 participants (mean age = 35.25, *SD* = 16.32; female = 116; male = 103; non-binary = 2) completed a third questionnaire again measuring their weekly consumption of meat and legumes. Finally, participants received feedback on the aims of the study. Figure 1 shows the flow of participants throughout the study.

### 3.2. Measures at Time 1

The questionnaire at T1 included several measures. Below, we report the measures relevant to the present paper.

*Identification with flexitarians.* Participants’ identification with flexitarians was assessed with seven items on a 7-point Likert scale ranging from “completely disagree” to “completely agree” (e.g., “If I think of the flexitarians… I feel like a member of this group”; adapted from [56]; *α* = 0.94). Higher values indicated a stronger identification with flexitarians.

*Attitude towards flexitarians.* Participants’ attitude towards flexitarians was assessed with three items using a 7-point semantic differential scale (e.g., “I admire flexitarians—flexitarians bother me”; adapted from [57]; *α* = 0.94). Higher values indicated a positive attitude towards flexitarians.

*Self-reported consumption of legumes.* Participants were asked to report their legume consumption over the previous week with three questions (“How many servings of legumes have you eaten in the previous week? Examples of legumes include beans, chickpeas, peas, lentils, broad beans, and lupin beans. An average portion of legumes is approximately 150 g for fresh or frozen legumes and 50 g for dried ones. A raw serving should be no more than the size and thickness of your palm”; adapted from [20]). Higher values indicated higher consumption of legumes.

*Self-reported meat consumption.* Participants were asked to report their consumption of red, processed, and white meat over the previous week (e.g., “How many servings of red meat have you eaten in the previous week? Red meat includes all meat that becomes dark when it is cooked and that is obtained from slaughter animals, such as veal, beef, or pork. A medium serving size for unprocessed red meat is about 100 g. A raw serving should be no more than the size and thickness of your palm” [20]). A sum score was calculated to obtain the total consumption of meat. Higher values indicated higher meat consumption.

### 3.3. Messaging Intervention

Every day for a period of two weeks (between T1 and T2), all participants received persuasive messages via the PsyMe app. Appendix A shows the full list of fourteen messages sent to participants as a function of the different message conditions.

Participants in the *addition message (ADD)* condition received messages focused on the low environmental impact of producing legumes (e.g., “Legumes’ cultivation requires limited use of fertilizers. Thus, it has a low pollution impact. If you eat legumes, you protect the planet from pollution”). Participants in the *replacement message (REP)* condition received messages focused on the lower environmental impact of cultivating legumes than producing meat (e.g., “Compared to livestock feeding production, legume cultivation requires a limited use of fertilizers. Thus, it has a lower pollution impact. If you eat legumes rather than meat, you protect the planet from pollution”). Participants in the *addition + dynamic norm message (ADD + DYN)* condition received the addition messages combined with information on the dynamic norm (e.g., “Nowadays, more and more people eat legumes. Legumes’ cultivation requires limited use of fertilizers. Thus, it has a low pollution impact. If you also eat legumes, you protect the planet from pollution”). Participants in the *replacement + dynamic norm message (REP + DYN)* condition received the replacement messages combined with a reference to the dynamic norm (e.g., “Nowadays, more and more people eat legumes instead of meat. Compared to livestock feeding production, legume cultivation requires limited use of fertilizers. Thus, it has a low pollution impact. If you also eat legumes instead of meat, you protect the planet from pollution”). Participants in the *control* condition did not receive any message.

### 3.4. Measures at Time 2

At T2, we again measured participants’ consumption of legumes and meat using the same scales employed at T1 (Table 1). We also measured participants’ views and readings of the messages. Then, we assessed the volunteers’ evaluation of the messages. Means and standard deviations of the variables at T2 are reported in Table 1.

Message views frequency was obtained through the PsyMe app, which keeps track of the reception of the messages.

Message reading frequency was assessed with an item rated on a scale ranging from “never” (1) to “always” (5) (“How many times did you read the messages?”).

Manipulation check was conducted by asking participants to select, among four messages, the one most similar to the messages they had received for fourteen days through the PsyMe app.

Message involvement was measured with three items using a 7-point Likert scale ranging from “completely disagree” to “completely agree” (e.g., “Messages involved me in what they had to say”; α = 0.85; [58]).

Message trust was assessed with three items on a 7-point Likert scale ranging from “not at all” to “completely” (e.g., “The information is credible”; α = 0.89; [58]).

Systematic processing was assessed with five items on a 7-point Likert scale ranging from “not at all” to “completely” (e.g., “While reading the messages, I thought about what actions I might take based on what I read”; α = 0.86; [22]).

### 3.5. Measures at Time 3

At T3, we again measured participants’ consumption of legumes and meat, using the same scales employed at T1 and at T2.

## 4. Results

### 4.1. Preliminary Analysis

All analyses were conducted in SPSS 23. Table 1 reports the means and SDs of the study variables at Times 1, 2, and 3.

First, we tested differences between conditions on study variables at T1 (identification with flexitarians, attitude towards flexitarians, consumption of legumes and meat, and age) with analyses of variance. The results did not show any significant main effect of message conditions on these variables (*p* > 0.10). Chi-square also did not show any significant differences in gender across different conditions (*p* = 0.19). These findings suggest that randomization was adequate, with the five conditions being comparable on the baseline variables.

Regarding dropouts (Figure 1), 129 participants dropped out at the post-test stage (T2). At T3, i.e., one month after the messaging intervention, we had 75 further dropouts. Chi-square did not show any significant difference in dropouts across conditions (*p* = 0.10). Then, we checked if the difference in the number of dropouts at T2 and T3 was determined by participants’ values in the study variables at T1 and/or their interaction with conditions. The results showed that dropouts at T2 did slightly differ in consumption of legumes at T1 (*p =* 0.002) but not in other variables (all *p >* 0.17). Precisely, dropouts at T2 had a marginally significant higher consumption of legumes (*M* = 2.33; *SD* = 1.57) than those who continued to participate (*M* = 1.69; *SD* = 1.32). However, there was neither an interaction between baseline legume consumption and conditions, nor a difference in dropouts at T3 compared to both those who did not drop and those who dropped at T2. Thus, we can assert that the final sample was representative of the initial sample.

### 4.2. Message Evaluation

A total of 49% of participants viewed all messages, 38.8% read from 7 to 13 messages, and the remaining 12.2% read at least 1 message. Moreover, 75% of participants reported that they always read the messages. As for the manipulation check, a chi-square analysis showed significant differences in message identification across different conditions (*χ^2^* = 196.17; *p* = 0.001). At the same time, message view and message reading frequencies were not influenced by conditions (all *p* > 0.24). All participants perceived the messages as involving and credible, and elaborated them thoroughly regardless of their content (*F*(3,161) = 1.36 *p* > 0.254*, ηp^2^* > 0.02).

### 4.3. Effects of Messages on Legume and Meat Consumption at Time 2

To answer our RQ1 and RQ2, we conducted a 5 (ADD, REP, ADD + DYN, REP + DYN, control condition) × 3 (T1 vs. T2 vs. T3) MANOVA with legume consumption and meat consumption as dependent variables, and with repeated measures on the last factor (Table 2). Multivariate results showed significant main effects of time and of interaction between time and message condition. The main effect of the condition was not significant. Univariate analyses showed a significant main effect of time on consumption of both legumes and meat. Pairwise comparisons (all *p* < 0.05) showed that participants consumed more legumes at T2 and T3 than at T1. Consistently, participants ate less meat at T2 and T3 than at T1. Univariate results also indicated a significant interaction effect between time and condition on consumption of both legumes and meat. Below we describe the significant outcomes of the pairwise comparisons according to message condition in detail (all *p* < 0.05).

Participants in the *control condition* did not report any change in legume and meat consumption at the end of the intervention (T2) and one month after (T3).

Participants in the *ADD message condition* reported a higher consumption of legumes at T2 compared to T1 and to participants in the control condition. However, at T3, these participants reported a lower consumption of legumes than T2. At follow up (T3), these participants reported a lower meat consumption than those in the control condition.

Participants in the *REP message condition* reported a marginally significant lower meat consumption at T2 compared to T1, and a significantly lower consumption of meat at T3 than control. Participants in the *ADD + DYN message condition* consumed more legumes at T3 than those in the control condition and less meat at T2 and T3 compared to T1.

At T2, participants in the *REP + DYN message condition* reported higher consumption of legumes compared to T1, and a greater consumption of legumes than control at T3. Compared to T1 and T2, at T3 they reported a lower meat consumption than control.

### 4.4. The Moderating Role of Identification with and Attitude towards Flexitarians at Time 2

To test which message content was more effective in changing participants’ food choices according to the levels of identification with flexitarians and attitude towards them (**H1–H3**), we calculated the change in participants’ consumption of legumes and meat as difference scores between the weekly consumption at T2 and T1. Then, we ran a moderation model with message condition as the independent variable, identification with flexitarians and attitude towards flexitarians as the moderators, and change in the consumption of legumes or meat as the dependent variables (Model 3 of the PROCESS macro for SPSS [59]). For the sake of simplicity, in this section, we discuss only the main results of this analysis. All findings are reported in Appendix B.

Regarding change in legume consumption (Appendix B—Table A2), we did not find any difference between conditions based on participants’ level of identification with flexitarians. Instead, attitude towards flexitarians had a significant interaction with the ADD condition (Figure 2). For participants with a positive attitude toward flexitarians, there was no conditional effect. Instead, participants with a negative attitude toward flexitarians increased their consumption of legumes more in the ADD condition than in the other condition. This result confirmed that messages on the addition of legumes to one’s diet to protect the environment are particularly effective when people have a negative evaluation of flexitarians.

We also found two significant three-way interactions (Figure 3): the interaction between the two moderators and the REP + DYN condition and the interaction between the two moderators and the ADD condition. In the case of participants with weak identification with flexitarians but a positive attitude towards them, the REP + DYN condition had the highest effect in increasing the consumption of legumes. Considering participants with weak or medium identification with flexitarians and a negative attitude towards them, the ADD condition had the highest effect. That is, participants with weak identification with flexitarians and a neutral attitude towards flexitarians or medium identification with flexitarians and a negative attitude toward flexitarians increased their consumption of legumes more when receiving addition messages than when receiving no messages.

Regarding change in meat consumption (Appendix B—Table A3), when controlling for moderators, we observed that not only ADD + DYN messages, but also REP messages decreased meat consumption. Moreover, there was a significant interaction between the REP condition and participants’ attitude towards flexitarians (Figure 4). Only participants who positively evaluated the flexitarians reduced meat consumption when receiving messages that recommended replacing meat with legumes to protect the environment. No other interaction was significant.

### 4.5. The Moderating Role of Identification with and Attitude towards Flexitarians at Time 3

We then calculated the change in legume and meat consumption at T3 as a difference score between the weekly consumption at T3 and T1 and conducted the same moderated analyses reported above. The moderation analysis on changes in consumption of legumes at T3 showed that ADD, ADD + DYN, and REP, REP + DYN conditions significantly interacted with participants’ identification with flexitarians, indicating that participants with a strong identification with flexitarians increased their consumption of legumes more than control (Appendix B—Table A4; Figure 5). However, we found neither the interaction between condition and attitude towards flexitarians nor the interaction between condition and both moderators.

A further moderation analysis on the change in meat consumption at T3 showed a reduction in meat at Time 3 in all message conditions but did not reveal any significant interaction between condition and moderators (Appendix B—Table A5).

## 5. Discussion

In the present study, we tested the impact of a 14-day messaging intervention promoting legume consumption. At the end of the two-week intervention (T2) and after one more month (T3), we observed a higher consumption of legumes and a lower consumption of meat, as compared to the self-reported intake at the beginning of the study (T1). Thus, we can assert that the message intervention was effective in promoting a shift towards a more plant-based diet. Below, we detail the effects of each message condition on the consumption of legumes and meat.

### 5.1. Short-Term Effects of Message Intervention on the Consumption of Legumes and Meat

Regarding legume consumption, at the end of the intervention, addition but not replacement messages increased participants’ intake of legumes compared to control. This finding suggests that people in an early stage of change prefer receiving messages recommending the inclusion of a new positive behavior (eating more legumes) rather than the modification of an established negative behavior (eating too much meat). This result is in line with the evidence of behavioral psychology, which has widely supported that it is easier to modify people’s habits by reinforcing positive behaviors rather than penalizing negative ones (i.e., differential reinforcement [60]).

Even if our main goal was changing people’s overall diet in the long term, we also considered short-term effects for two main reasons. First, previous studies on the topic mainly evaluated short-term effects, while only a few considered long-term effects. Therefore, considering the short-term outcomes allowed us to compare our results with previous literature on the topic. Second, the short-term effects still indicate the potential of the messages. The long-term effects were measured when the messages were no longer sent. Therefore, we cannot exclude that messages that were effective in the short term could have been even more effective if sent for a longer time.

*Addition messages* were especially effective in increasing the consumption of legumes at T2 when participants had a weak identification with flexitarians and evaluated them negatively. Probably, people with such negative predispositions towards flexitarians can accept adding legumes as a food choice consistent with their dietary identity as omnivores. On the contrary, they perceive replacement recommendations as the proposal of a behavior that is implemented by people they perceive as different from them, and they are resistant to behave accordingly. Similarly, those who have negative attitudes towards flexitarians can perceive the information that others are changing their diet to protect the environment (i.e., the inclusion of a reference to a dynamic norm in the addition or replacement message) as implying the moral superiority of being flexitarians. In turn, this perceived moral superiority can make participants reactant [61] and motivate them to protect their sense of morality and freedom from being influenced [62].

Finally, the addition message did not reduce meat consumption at T2. Therefore, at first, participants receiving messages on the addition of legumes to their diet followed this recommendation without necessarily considering the consumption of legumes as a substitute for meat. Future studies could investigate whether at this stage people add legumes without changing their meat choices or if they do so by substituting other food products that were not assessed in this research.

As for the short-term effectiveness of *replacement messages*, they reduced meat consumption at T2 especially when participants had a positive attitude towards flexitarians. Probably, in the replacement message condition, people recognized substituting meat to protect the environment as a typical flexitarian behavior. Thus, they were more willing to adopt this positively evaluated behavior. This is in line with past research suggesting that a positive attitude towards a minority group position may increase people’s motivation to adopt the group goals as personal goals [63].

The *addition + dynamic norm message* was effective in increasing the portions of legumes eaten in the last week of the intervention, even if there was not a significant difference with control at T2. This suggests that adding a reference to dynamic norms to the addition message is not effective enough in the early stages of change.

The *replacement + dynamic norm message* was indeed effective in increasing legume consumption, especially when participants had a positive attitude towards flexitarians and a weak identification with them. This result suggests that these people need to receive the recommendation to replace meat with legumes combined with the information that others are succeeding in adopting a flexitarian diet. In other words, knowing that other people are succeeding in increasing the consumption of legumes can help people with a positive evaluation of the flexitarian group and an acknowledgement that they are not yet part of it to adopt such a minority behavior goal.

### 5.2. Long-Term Effects of Message Intervention on the Consumption of Legumes and Meat

One month after the end of the intervention, the effects of the messages on participants’ consumption of legumes and meat differed from those observed immediately after the end of the intervention.

As for the consumption of legumes, *addition messages* were not effective. After an initial increase in the consumption of legumes, those who did not have a strong identification with flexitarians returned to the initial consumption levels. The addition recommendation needs to be matched with a reference to dynamic norms to be effective in the long run. The *addition + dynamic norm message* helps people to continue increasing their consumption of legumes after the end of the intervention, even more when people have a strong identification with flexitarians.

Even if not increasing legume consumption in the short term, *replacement messages* reached this goal in the long term, but again only when this dietary recommendation was matched with a reference to dynamic norms. Like addition + dynamic norm messages, *replacement + dynamic norm messages* supported people to continue adding legumes to their diet after the end of the intervention, more so when having a strong identification with flexitarians.

As for the effect of intervention on meat consumption, at the end of the intervention, we observed only a marginal reduction in meat consumption among people receiving replacement messages. One month after the end of the intervention, all participants who had received the messages reduced their consumption of meat, compared to those who had not received them. Environmental messages were therefore effective only after people had taken time to internalize the proposed recommendations, regardless of the type of eating strategy (i.e., addition or replacement) suggested in the message or the reference to the behavior of others (i.e., information on dynamic norm). However, the most effective message intervention was the *replacement + dynamic norm intervention*.

These findings suggest that dynamic norms have a stronger effect in the long term than in the short term. Apparently, a dietary change to adhere to a (still) minority norm can only take place in the long term, after a careful analysis of what the minority represents for the individual. If portraying a behavior as increasing in popularity can indeed spur compliance even to minority norms [64], social minorities influence people’s behaviors through a validation process that elicits closer attention to, and deeper elaboration of, the minority views [65]. Hence, people do not feel pressured to conform to minority positions, and only endorse them as the result of a true influence process. The fact that the observed change takes place through an in-depth elaboration of what is recommended therefore cancels the effect of the type of food strategy suggested and facilitates the maintenance of the behavior over time.

Another interesting aspect that emerged in the long term is the role of identification with flexitarians. Identifying with flexitarians does not influence the immediate effects of exposure to messages, as we observed an effectiveness of addition and replacement messages at T2 regardless of how much participants identified with flexitarians. However, identification plays an important role in the long run. The more participants identified themselves with the flexitarians, the more they increased legume consumption at T3 in all message conditions.

In light of the present results, the lack of a behavioral effect observed in past studies may be due to participants reading the messages only once. People likely need more frequent exposure to environmental arguments to initiate a behavioral change that requires ongoing commitment over time, as demonstrated by the effectiveness of longer messaging interventions focused on reducing meat consumption (e.g., [40]).

### 5.3. Theoretical and Practical Implications

The above results advance our comprehension of how to use environmental messages aimed at promoting a shift toward a more plant-based diet.

First, the recommendation to adopt an *addition dietary strategy* to protect the environment is effective in increasing legume consumption in the short term. However, in the long term, this recommendation is effective only when combined with reference to a dynamic norm. Second, the recommendation to adopt a *replacement dietary strategy* to protect the environment is effective in reducing meat consumption, but only when people evaluate being flexitarian positively. Again, in the long term, the replacement messages are more effective when combined with reference to a dynamic norm.

In a previous study on the promotion of plant-based meat, not only addition but also replacement messages were immediately effective [30]. We can interpret the difference between these findings and the present ones by referring to two main considerations. On one hand, addition and replacement messages can have a differential effect based on the intrinsic attribute of the recommended food. Although both the present and the previous study presented the addition of legumes and plant-based meat as a pro-environmental food choice, consumers may have not perceived them as similar recommendations. Unlike eating legumes, eating plant-based meat is likely to be immediately associated with the goal of meat replacement, as the label “plant-based meat” itself suggests. On the other hand, people can perceive the production of legumes as having evident and immediate environmental benefits (e.g., crop rotation and biological nitrogen fixation). Instead, the manufacturing of plant-based meat has less evident and more indirect environmental benefits. Since it involves a relevant expenditure of environmental resources for its manufacturing, its consumption is only justified as an alternative to meat, whose production has an even greater environmental impact. These results suggest that future studies should usefully consider whether the effectiveness of a food strategy focused on in promoting environmental protection varies as a function of the type of food on which the communicative strategy.

Second, our study showed that the effect of reference to dynamic norms in recommendation messages is not always immediate. On the contrary, it can take a long time, at least when, as in this case, the behavior to be implemented is difficult. Future studies will be able to better investigate the effect of this type of social influence over longer periods, also trying to understand why their effect can be so delayed. Despite this, the results of our study recommend including this type of descriptive norm in communications aimed at promoting a sustainable diet.

Finally, the results of our study open new ways to use public communication campaigns tailored to the receivers’ psychological characteristics. Promotional campaigns might adopt personalized communication using chatbots or smartphone applications that allow assessing receivers’ attitudes and identity patterns and sending tailored messages based on them. As we have seen, the addition messages are suitable for individuals with negative attitude towards flexitarians and weak identification with them, while the replacement + dynamic norm messages are the preferable messages for individuals with positive attitude towards flexitarians and weak identification with them.

### 5.4. Study Limitations

This research has some limitations that future studies might address. First, to avoid the possible influence of social desirability and memory biases linked to self-reported behaviors, future research may confirm the effectiveness of our messages by measuring their impact on actual behavior, for example, by asking participants to fill in daily food diaries. Second, as our sample was composed only of Italian people and was not fully balanced in terms of some sociodemographic variables (e.g., education and age), future scholars may control for the level of generalizability of our results. Third, checking whether the effectiveness of this intervention is maintained over a longer period would be useful. A longer study could also verify if the changes in the consumption of legumes and meat will modify people’s attitudes towards flexitarians and their identification with them. Fourth, this study mainly captured the “rational” processes of message evaluation and impact, overlooking the possibility that emotional reactions (such as fear or regret) or moral considerations (i.e., ascription of responsibility) might mediate the message effects [27,66,67,68,69,70,71].

## 6. Conclusions

The findings of this study suggest that inviting people to add legumes to their diet is the best solution for immediately increasing this food choice, especially in those who can be resistant to adopting a flexitarian diet. Messages encouraging the replacement of meat with legumes are the best option to reduce meat consumption for those who have positive attitudes towards flexitarians. However, only the reference to the dynamic norm induced people to eat more legumes after the end of message sending, and this happened regardless of the type of behavioral strategy suggested. Therefore, reference to a dynamic norm can take longer for a behavioral change to occur but has a better chance of making it long lasting.

## Figures and Tables

**Figure 1 nutrients-15-00015-f001:**
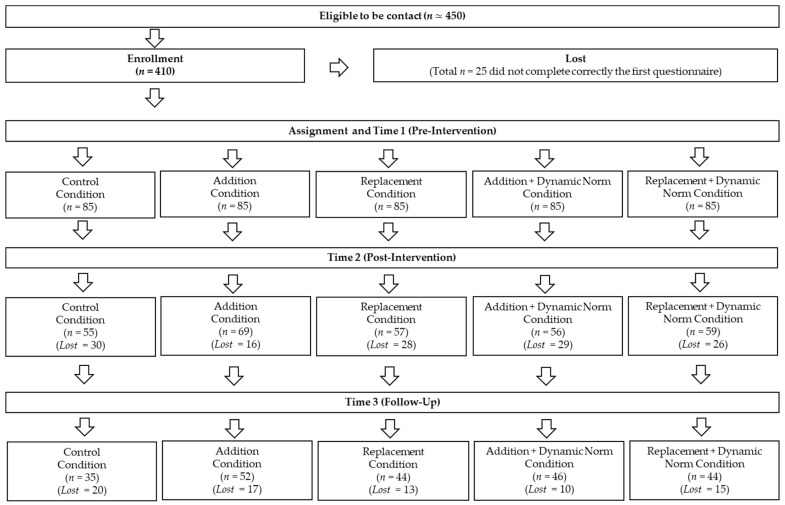
Flowchart of participant recruitment.

**Figure 2 nutrients-15-00015-f002:**
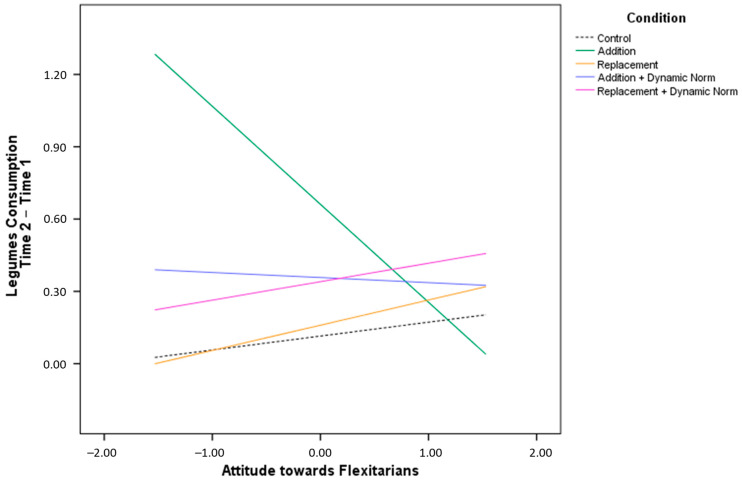
Changes in legume consumption at Time 2 minus Time 1 based on participants’ attitude towards flexitarians.

**Figure 3 nutrients-15-00015-f003:**
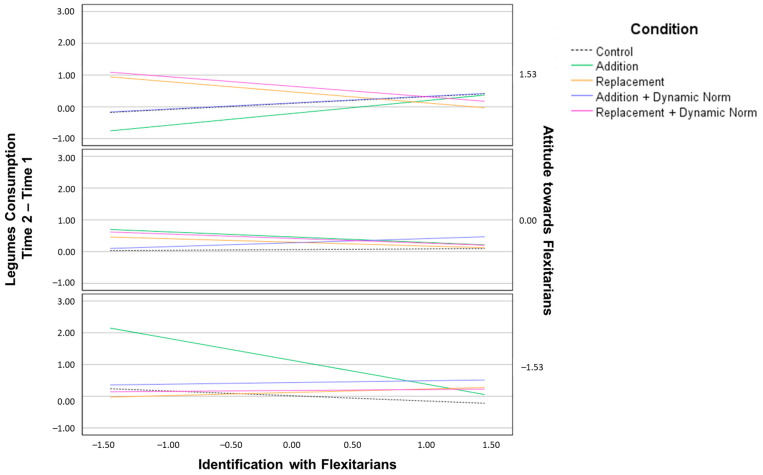
Changes in legume consumption at Time 2 minus Time 1 based on participants’ attitude towards flexitarians and identification with them.

**Figure 4 nutrients-15-00015-f004:**
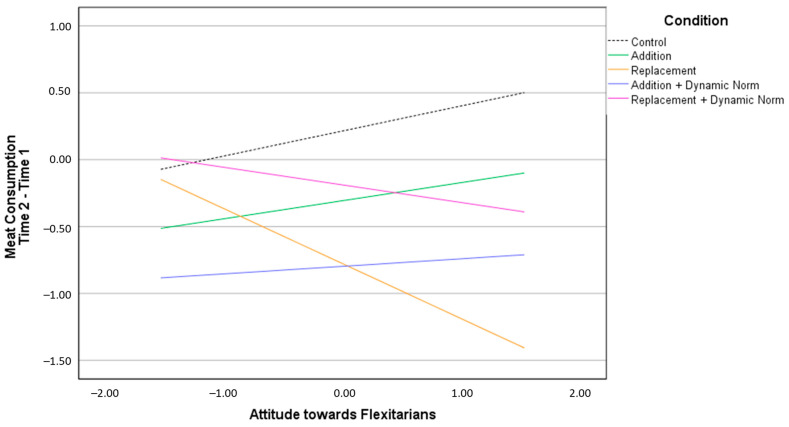
Changes in meat consumption at Time 2 based on participants’ attitude towards flexitarians.

**Figure 5 nutrients-15-00015-f005:**
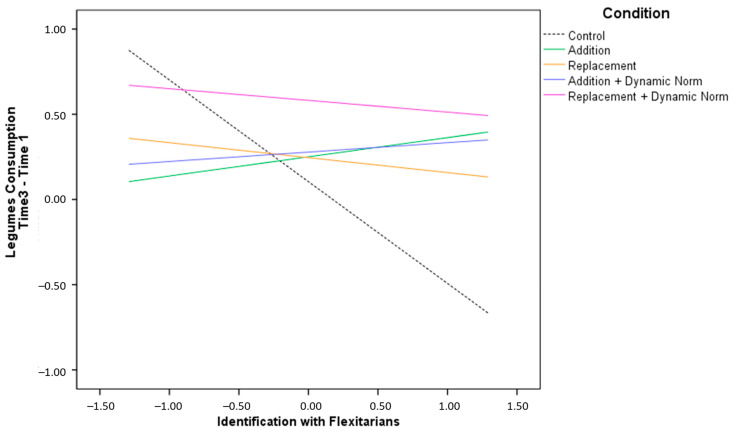
Changes in meat consumption at Time 2 based on participants’ identification with flexitarians.

**Table 1 nutrients-15-00015-t001:** Means and standard deviations of dependent variables in each condition at Time 1 (baseline), Time 2 (post-intervention), and Time 3 (follow-up).

	Control Condition			Addition Condition			Replacement Condition			Addition + Dynamic Norm Condition			Replacement + Dynamic Norm Condition		
	Time 1	Time 2	Time 3	Time 1	Time 2	Time 3	Time 1	Time 2	Time 3	Time 1	Time 2	Time 3	Time 1	Time 2	Time 3
	*M* *(SD)*	*M* *(SD)*	*M* *(SD)*	*M* *(SD)*	*M* *(SD)*	*M* *(SD)*	*M* *(SD)*	*M* *(SD)*	*M* *(SD)*	*M* *(SD)*	*M* *(SD)*	*M* *(SD)*	*M* *(SD)*	*M* *(SD)*	*M* *(SD)*
Identification with Flexitarians	2.81 (1.26)	-	-	2.65 (1.32)	-	-	2.92 (1.29)	-	-	3.02 (1.34)	-	-	2.82 (1.39)	-	-
Intergroup Attitude towards Flexitarians	5.29 (1.30)	-	-	5.15 (1.50)	-	-	4.95 (1.61)	-	-	5.24 (1.44)	-	-	5.13 (1.48)	-	-
Message View Frequency	-	-	-	-	10.84 (4.09)	-	-	11.68 (3.46)	-	-	11.94 (3.23)	-	-	11.05 (3.75)	-
Message Readings Frequency	-	-	-	-	3.63 (0.72)	-	-	3.73 (0.67)	-	-	3.71 (0.62)	-	-	3.59 (0.71)	-
Message Involvement	-	-	-	-	5.00 (1.28)	-	-	5.12 (0.94)	-	-	5.16 (1.12)	-	-	5.03 (1.33)	-
Message Trust	-	-	-	-	4.90 (0.93)	-	-	4.88 (0.77)	-	-	5.00 (0.90)	-	-	4.72 (0.89)	-
Systematic Processing	-	-	-	-	4.99 (0.85)	-	-	4.75 (1.19)	-	-	4.88 (0.93)	-	-	4.98 (1.25)	
Legume Consumption	1.84 (1.36)	1.84 (1.24)	1.51 (0.85)	1.89 (1.53)	2.46 (2.08)	1.77 (1.50)	1.85 (1.35)	2.05 (1.71)	1.98 (1.09)	1.91 (1.42)	2.27 (1.49)	2.26 (2.02)	1.86 (1.32)	2.22 (1.23)	2.16 (1.27)
Meat Consumption	6.78 (2.32)	6.80 (2.62)	7.43 (2.74)	6.77 (1.99)	6.49 (2.24)	6.19 (2.38)	6.70 (2.48)	6.00 (2.66)	6.16 (2.58)	6.90 (2.13)	5.82 (2.43)	5.65 (2.76)	6.71 (2.05)	6.37 (2.87)	5.48 (2.59)

**Table 2 nutrients-15-00015-t002:** Results of repeated measures multivariate analysis of variance to test the impact of message intervention on self-reported consumption of legumes and meat, both in the short term (i.e., Time 2) and in the long term (Time 3).

	*df*	*F*	*p*	*ηp2*
Multivariate Effects				
Time	4213	5.77	0.001	0.10
Time X Message Condition	16,864	2.73	0.001	0.05
Message Condition	8432	1.01	0.42	0.02
Univariate Effects on Legume Consumption				
Time	2432	6.34	0.002	0.03
Time X Message Condition	8432	1.93	0.05	0.03
Univariate Effects on Meat Consumption				
Time	2432	5.37	0.005	0.03
Time X Message Condition	8432	3.82	0.001	0.07

## Data Availability

The data presented in this study are available on request from the corresponding author.

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
