# Peer review of "Legumes or Meat? The Effectiveness of Recommendation Messages towards a Plant-Based Diet Depends on People’s Identification with Flexitarians"

_nutrients, 2022, doi:10.3390/nu15010015_

Round 1
Reviewer 1 Report
The study aimed to compare the promotion of a plant-based diet using a 2-wk mobile app intervention in which messages either emphasizing the addition of legumes or emphasizing the replacement of meat with legumes. In addition, the messages were either combined (or not) with dynamic norms with the assumption that dynamic norms would also influence actual consumption behavior. Short- and long-term effects were assessed. The results show the effectiveness of addition messages in the short-term, particularly in people that have a low identification with flexitarians. In the long-term, legume consumption was only increased in the group with addition messages combined with dynamic norm. Meat consumption was reduced in the replacement message condition in people that have a positive attitude towards flexitarians.
The topic is significant for the nutrition community given the global importance to eat a more plant-based diet and to change people’s food intake. However, the manuscript is way too long and looses the reader halfway through. Both, the introduction and the results need to be shortened considerably. Particularly the results need a better structure to help the reader navigate through the findings. Some parts can also be moved to the discussion section (e.g. page 10, line 439ff).
Minor comments:
Page 2, line 71. If previous findings are controversial why do the authors conclude “that environmental messages are only persuasive when manipulated in certain ways”?
Page 2, line 81ff. The authors point out that only one study compared the three different dietary strategies to change one’s diet and give this as a reason for choosing their two message strategies. Shouldn’t the design then have incorporated three strategies if that was your reasoning? Why was one strategy left out? (In general, this paragraph can be shortened). And why did only the replacement message work for people who already ate plant-based meat? And on page 3, line 122, the authors mention a study with two different dietary strategies, again, it is not clear why the authors used two instead of three strategies.
Page 4, line 156ff, this sentence needs a reference.
Page 5, line 227, why would it be interesting to look at short-term changes given the topic? Why would it be interesting to know short-term effects when the goal is to change people’s overall diet in the long term?
Page 5, 3. Method, which variable was used to calculate the power for this study?
Page 6, line 281ff. Were examples for servings given? How did the authors ensure that the participants know what a serving of different legumes or meat looks like?
Page 7, Table. Footnotes indicating the measuring units are necessary (or need to be added to the first column).
Figure 2+3 seem to miss words in the captions.
Page 13/14, line 472ff. The end of the sentence seems to be missing.
Page 14, line 530. This does not sound like a proper English sentence.
Author Response
December 15, 2022
Dear Reviewer,
We thank you for the feedback on our paper. In revising it, we have addressed all the concerns that you provided for our manuscript. We hope the new version will be suitable for publication in Nutrients.
Please, find below a detailed response to your points, with the original comments in italics.
Thank you for your revision.
Best regards,
The Authors
- The study aimed to compare the promotion of a plant-based diet using a 2-wk mobile app intervention in which messages either emphasizing the addition of legumes or emphasizing the replacement of meat with legumes. In addition, the messages were either combined (or not) with dynamic norms with the assumption that dynamic norms would also influence actual consumption behavior. Short- and long-term effects were assessed. The results show the effectiveness of addition messages in the short-term, particularly in people that have a low identification with flexitarians. In the long-term, legume consumption was only increased in the group with addition messages combined with dynamic norm. Meat consumption was reduced in the replacement message condition in people that have a positive attitude towards flexitarians. The topic is significant for the nutrition community given the global importance to eat a more plant-based diet and to change people’s food intake. However, the manuscript is way too long and looses the reader halfway through. Both, the introduction and the results need to be shortened considerably. Particularly the results need a better structure to help the reader navigate through the findings. Some parts can also be moved to the discussion section (e.g. page 10, line 439ff).
- Thanks for your suggestion. In the revised version, we have now shortened the Introduction (lines 26-65), Theoretical Framework (lines 67-251), and Results sections (lines 354-561) of the paper. We hope this has made the paper more readable.
Minor comments:
- Page 2, line 71. If previous findings are controversial why do the authors conclude “that environmental messages are only persuasive when manipulated in certain ways”?
- We have now reformulated this sentence explaining what factors can determine the persuasiveness of the messages (lines 77-112).
- Page 2, line 81ff. The authors point out that only one study compared the three different dietary strategies to change one’s diet and give this as a reason for choosing their two message strategies. Shouldn’t the design then have incorporated three strategies if that was your reasoning? Why was one strategy left out? (In general, this paragraph can be shortened). And why did only the replacement message work for people who already ate plant-based meat? And on page 3, line 122, the authors mention a study with two different dietary strategies, again, it is not clear why the authors used two instead of three strategies.
- We have now clarified why only two of the three strategies were considered in the two researches (lines 93-95). We have also briefly clarified the explanation of the results obtained in the first research cited (lines 99-100). We have also shortened this paragraph.
- Page 4, line 156ff, this sentence needs a reference.
- While shortening this paragraph, we have now deleted this sentence.
- Page 5, line 227, why would it be interesting to look at short-term changes given the topic? Why would it be interesting to know short-term effects when the goal is to change people’s overall diet in the long term?
- Thank you for giving us the opportunity to better clarify this aspect in the paper. In our analyses, we have considered short-term effects for two main reasons. First, previous studies on the topic evaluated short-term effects, while only a few considered long-term effects. Therefore, considering the short-term outcomes allowed us comparing our findings with previous literature on the subject. Second, the short-term effects still indicate a potential of the messages. The long-term effects were measured when the messages were no longer sent. Therefore, we cannot exclude that the messages that were effective in the short term might have been even more effective if sent for a longer time. We have now discussed these points in the Discussion section (lines 591-598).
- Page 5, 3. Method, which variable was used to calculate the power for this study?
- We have now clarified that the variables used were the self-reported consumption of meat and the self-reported consumption of legumes.
- Page 6, line 281ff. Were examples for servings given? How did the authors ensure that the participants know what a serving of different legumes or meat looks like?
- We have now reported more details on the instructions we provided to participants regarding how to measure an average portion of meat or legumes (lines 292-305).
- Page 7, Table. Footnotes indicating the measuring units are necessary (or need to be added to the first column).
- We have now added footnotes to this table.
- Figure 2+3 seem to miss words in the captions.
- We have now corrected these captions.
- Page 13/14, line 472ff. The end of the sentence seems to be missing.
- Sorry, but we were unable to trace this error. However, we have found a mistake of this type elsewhere in the text and corrected it (710-712).
- Page 14, line 530. This does not sound like a proper English sentence.
- We have now deleted this sentence.
Reviewer 2 Report
The paper is well structured and addresses an interesting and valuable issue. It can be improved with the following suggestions.
Introduction. This section should include a more specific part on the aim and objectives of the paper.
Literature. The author(s) should emphasize that the study focuses on environmental text messages. Literature section does not include a reference to other types of messages, for example health messages, or image-based messages. Moreover, it does not mention potential consumers' emotional responses.
Methodology: The paper does not consider any exploratory or pilot study and it is not indicated how the author/s developed the messages used in the experimental conditions.
Titles of Figure 2 and Figure 3 are not correct. There are two Figures 3.
There are some typing errors and references do not adhere to the journal guidelines.
Author Response
December 15, 2022
Dear Reviewer,
We thank you for the positive feedback on our paper. In revising it, we have addressed all the concerns that you provided for our manuscript. We hope the new version will be suitable for publication in Nutrients.
Please, find below a detailed response to your points, with the original comments in italics.
Thank you for your revision.
Best regards,
The Authors
- The paper is well structured and addresses an interesting and valuable issue. It can be improved with the following suggestions.
Introduction. This section should include a more specific part on the aim and objectives of the paper.
- We have now better clarified the aim and the objectives of the paper at the end of the Introduction (lines 53-64).
- The author(s) should emphasize that the study focuses on environmental text messages. Literature section does not include a reference to other types of messages, for example health messages, or image-based messages.
- As suggested we have now included a reference to other types of messages and clarified why we focused only on the environmental ones (lines 67-74).
- Moreover, it does not mention potential consumers' emotional responses.
- In the Study Limitation section, we have now discussed the importance of considering emotional responses in future studies (lines 739-742).
- Methodology: The paper does not consider any exploratory or pilot study and it is not indicated how the author/s developed the messages used in the experimental conditions.
- Although we have not conducted a pilot study of the messages, we used a manipulation check to assess whether participants properly understood all messages (lines 382-394). In addition, all participants reported having perceived the messages as credible and involving (lines 382-394).
- Titles of Figure 3 and Figure 4 are not correct. There are two Figures 4.
- We have now corrected the numerical ordering of the figures.
- There are some typing errors and references do not adhere to the journal guidelines.
- We have now checked all typos and references according to the journal guidelines.